# Melanoma-Derived Exosomal miR-125b-5p Educates Tumor Associated Macrophages (TAMs) by Targeting Lysosomal Acid Lipase A (LIPA)

**DOI:** 10.3390/cancers12020464

**Published:** 2020-02-17

**Authors:** Dennis Gerloff, Jana Lützkendorf, Rose K.C. Moritz, Tom Wersig, Karsten Mäder, Lutz P. Müller, Cord Sunderkötter

**Affiliations:** 1Department of Dermatology and Venereology, Martin-Luther-University Halle-Wittenberg, 06120 Halle (Saale), Germanycord.sunderkoetter@uk-halle.de (C.S.); 2Department of Internal Medicine IV, Hematology and Oncology, Martin-Luther-University Halle-Wittenberg, 06120 Halle (Saale), Germany; jana.luetzkendorf@uk-halle.de (J.L.); lutz.mueller@uk-halle.de (L.P.M.); 3Institute of Pharmacy, Faculty of Biosciences, Martin Luther University Halle-Wittenberg, 06120 Halle (Saale), Germany; tom.wersig@pharmazie.uni-halle.de (T.W.); karsten.maeder@pharmazie.uni-halle.de (K.M.)

**Keywords:** exosomes, miRNAs, tumor-associated macrophages, melanoma

## Abstract

Tumor-associated macrophages (TAMs) are the most abundant immune cells in the tumor microenvironment, promoting tumor initiation, growth, progression, metastasis, and immune evasion. Recently it was shown that cancer cell-derived exosomes induce a tumor-promoting phenotype in TAMs. Exosome-loaded proteins, DNA, and RNAs may contribute to the macrophage reprogramming. However, the exact mediators and mechanisms, particularly in melanoma, are not known. In this study we examined the effects of cutaneous melanoma-derived exosomes on macrophage function and the underlying mechanisms. First, we showed that exposure to melanoma exosomes induces a tumor-promoting TAM phenotype in macrophages. Sequencing revealed enrichment for several miRNAs including miR-125b-5p in cutaneous melanoma exosomes. We showed that miR-125b-5p is delivered to macrophages by melanoma exosomes and partially induces the observed tumor-promoting TAM phenotype. Finally, we showed that miR-125b-5p targets the lysosomal acid lipase A (LIPA) in macrophages, which in turn contributes to their phenotype switch and promotes macrophage survival. Thus, our data show for the first time that miR-125b-5p transferred by cutaneous melanoma-derived exosomes induces a tumor-promoting TAM phenotype in macrophages.

## 1. Introduction

Exosomes are small cell-derived membrane nanovesicles of about 50–200 nm in diameter, loaded with proteins, DNA, and coding and non-coding RNAs, e.g., miRNAs [1,2]. They are released in the extracellular environment by living cells and act as carriers for intracellular cell-cell communication [3]. Cancer cell-derived exosomes differ from exosomes secreted by normal cells with regard to their cargo. Distinctive differences in content are meanwhile used as diagnostic or prognostic markers in some cancers, e.g., miR-21 is a well-known oncomiR, used as a diagnostic marker in ovarian and pancreatic cancer [4]. Recent studies show that exosomes released by cancer cells interact amongst others with tumor-infiltrating T cells (TILs) [5], myeloid-derived suppressor cells (MDSCs) or tumor-associated macrophages (TAMs) [6]. They induce a phenotypic switch of tumor-infiltrating immune cells and stroma cells, thereby creating a tumor-permissive microenvironment [7,8,9].

While TAMs are differentiated macrophages found in the microenvironment of a majority of malignant tumors, MDSCs comprise a heterogeneous population of myeloid progenitors, immature granulocytes, monocytes, and dendritic cells at different stages of differentiation, which also accumulate in the blood, bone marrow, and secondary lymphoid organs of tumor-bearing mice and men [10].

TAMs are found in high numbers in early melanoma lesions and have been shown to release a variety of cytokines and chemokines contributing to tumor growth, progression, metastasis, and immune evasion [11,12,13,14]. An infiltration of macrophages was associated with melanoma invasiveness, metastasis, and angiogenesis [15,16].

TAMs are an attractive target for therapeutic strategies aimed at reprogramming their pro-tumor phenotype to generate effective antitumor activity.

Macrophages, traditionally, are assigned as either classically activated M1 polarized or alternatively activated M2 polarized phenotypes. M1 macrophages are induced in vitro by INFγ and lipopolysaccharide (LPS) and are described as tumor suppressive, whereas M2 macrophages are stimulated by IL-4 and IL-13 and supposed to be tumorigenic. However, this strict distinction is not uniformly observed in malignancies in vivo, because TAMs are plastic and exhibit properties of both macrophage phenotypes [17].

A putative mechanism by which exosomes modify macrophage phenotypes includes the transfer of miRNAs, as miRNAs are involved in the regulation of immune cell phenotype and function [18]. In selected tumor models such as ovarian cancer, specific exosome-transported miRNAs were demonstrated to induce a tumor-promoting phenotype in macrophages [19,20]. miRNAs are small non-coding RNAs, that post-transcriptionally regulate protein expression by binding to the 3’ untranslated region (3’UTR) of target mRNAs or by promoting the degradation of mRNA [21,22]. They have the potential to regulate several target mRNAs at the same time. miRNAs have regulatory functions in a variety of processes, such as embryonal development [23], metabolism [24], and cell proliferation, [25] as well as disease development and progression [26].

We therefore wondered if melanoma cell-derived exosomes would induce a tumor-promoting phenotype in macrophages and, if so, whether they specifically contain certain miRNAs targeting the pathways which promote such a phenotypic change. We found that miR-125b-5p in the exosomes of melanoma cells acts as an important modulator of melanoma-associated macrophages.

## 2. Results

### 2.1. Isolation and Characterization of Exosomes Derived by Malignant Melanoma Cells

As exosomes derived from malignant cells have been shown to modify cells of the tumor microenvironment, we wanted to investigate the potential of melanoma-derived exosomes to reprogram tumor-associated macrophages. Given the lack of data on the impact of melanoma exosomes on the switch of the macrophage phenotype, we relied on established cell line models to exclude interindividual bias. To this end, we isolated exosomes released by melanoma cell lines from a 48 h conditioned medium and confirmed their characteristic features, such as the size distribution of approximately 160 nm, by nanoparticle tracking analysis (NTA) (Figure 1A). In order to evaluate the quality of exosome isolation we performed western blot and could confirm the expression of the exosomal markers CD63 and CD81 and excluded contamination from cellular components by analysis of the endoplasmatic reticulum protein calnexin (Figure 1B). To prove the functionality of isolated exosomes, we analyzed the uptake of SYTO-stained exosomes by M1 polarized macrophages. We demonstrated by microscopy and flow cytometry that THP-1-derived macrophages are capable of taking up melanoma-derived exosomes (Figure 1C).

### 2.2. Melanoma Exosomes Induce a Proinflammatory and Proangiogenic Phenotype Macrophages

As melanomas have been shown to harbor TAMs in their microenvironment, we wanted to investigate whether exosomes from melanoma cells are able to reprogram M1 macrophages into tumor-promoting TAMs.

Therefore, we co-cultured M1 polarized macrophages with MV3 melanoma cells or with exosomes, isolated from a MV3 conditioned medium for 48 h, and performed next-generation sequencing (NGS) expression analysis (Figure 2A). We found that the gene expression profiles of M1 macrophages treated with melanoma-derived exosomes or co-cultured with melanoma cells showed partially similar changes compared to M1 control macrophages. The gene ontology (GO) analyses of M1 macrophages incubated with melanoma exosomes revealed an enrichment of pathways involved in inflammation, angiogenesis, migration, and chemotaxis (Figure 2B). These results were validated by gene set enrichment analysis (GSEA) (Figure 2C). To confirm our findings, we performed quantitative real-time PCR (qRT-PCR) of THP-1-derived M1 macrophages, either after 48 h of co-culture with melanoma cell line MV3 or treated for 48 h with exosomes isolated from the supernatants of three different melanoma cell lines (MV3, WM35, and WM902B). We demonstrated a strong increased expression of the proinflammatory cytokines IL-6, TNFα, and IL-1β; proangiogenic factors IL-8 and VEGFa; as well as genes involved in chemotaxis and the recruitment of myeloid cells, such as CCL1 and CCL2 (Figure 2D). In addition, we analyzed the expression of further M1/M2 macrophage marker genes in M1 macrophages treated with exosomes derived from different melanoma cell lines by qRT-PCR and compared it to the basal expression in M1 and M2 polarized macrophages. The exosome-treated macrophages showed strong differences in the analyzed gene expression profiles towards M1/M2 macrophages. However, they showed a strong increased expression of M1-associated genes, while M2 marker genes were mostly downregulated (Appendix A). Furthermore, we observed morphological changes of exosome-treated M1 macrophages (Figure 2E)—their fraction contained a higher number of large, cytoplasm-rich macrophages, whereas the control cells showed more spindle-shaped cells. The observed morphological changes and the induction of gene expression increased with treatment time and rising exosome concentrations (Appendix A)

These results show that melanoma-derived exosomes have a strong impact on the phenotype of macrophages.

### 2.3. miR-125b-5p is Enriched in Melanoma-Derived Exosomes

For a better understanding of the mechanism leading to the exosomal-mediated phenotypical change of M1 macrophages we analyzed the cargo of exosomes. As miRNAs are major factors in post-transcriptional gene regulation and can rapidly regulate several target mRNAs, we investigated the miRNA content of exosomes released by normal melanocytes or by melanoma cell lines. Exosomal RNA was isolated from the normal human epidermal melanocytes (NHEM) of three different donors and from three different melanoma cell lines (WM9, WM35, and WM902B). miRNAs were analyzed according to a small RNA-specific next-generation sequencing protocol. When compared to NHEM-derived exosomes, we found 12 significantly enriched and 22 significantly decreased miRNAs in exosomes from the three melanoma cell lines (Figure 3A). Pathway analysis of the top five abundant and significantly enriched miRNAs in melanoma exosomes (hsa-miR-100-5p, hsa-miR-99b-5p, hsa-miR-221-3p, hsa-miR-24-3p, and hsa-miR-125b-5p) showed a participation in various cancer pathways, including melanoma (Figure 3B). The most abundant and significantly enriched miRNA in exosomes derived from melanoma cells was miR-125b-5p (Figure 3C). Its expression was also enriched in total melanoma cells compared to normal melanocytes (Appendix A). We subsequently confirmed this enrichment of miR-125b-5p in melanoma exosomes by qRT-PCR (Figure 3D).

### 2.4. miR-125b-5p is Delivered to M1 Macrophages by Melanoma Exosomes

In order to investigate if miR-125b-5p is delivered into melanoma-associated macrophages by exosomes, we analysed the expression of miR-125b-5p in untreated M1 polarized macrophages as well as in M1 polarized macrophages that had been treated with melanoma-derived exosomes. qRT-PCR analyses showed increased miR-125b-5p levels in exosome-treated M1 macrophages (Figure 4A). To exclude the endogenous induction of miR-125b-5p in M1 macrophages, we analysed the expression of the primary transcripts pri-miR-125b-1, pri-miR-125b-2, and the mature miR-125b-5p. In co-cultured as well as exosome-treated M1 macrophages, we did not observe a rise of the primary transcripts pri-miR-125b-1 and pri-miR-125b-2, whereas mature miR-125b-5p was enriched (Figure 4B,C). This allows the conclusion that mature miR-125b-5p was delivered into M1 macrophages by melanoma cell-released exosomes and was not the result of endogenous expression induction.

### 2.5. Overexpression of miR-125b-5p Partially Mimics Exosome-Induced Switch of TAM Phenotype and Promotes Survival of Macrophages

To test our hypothesis that miR-125b-5p participates in macrophage polarization, we transfected M1 macrophages with miR-125b-5p mimics or negative control. After 48 h we analyzed the gene expression and morphology. Using qRT-PCR we observed an increased expression of IL-1β, CCL1, CCL2, and CD80 (Figure 5A) in transfected M1 macrophages, similarly to in M1 macrophages co-cultured with melanoma cells or treated with melanoma-derived exosomes. However, we did not find a change in the expression of IL-6, TNFα, VEGFA, and IL-8 after miR-125b-5p overexpression (Appendix A). Morphological analyses indicated a phenotypical change of M1 macrophages after transfection with miR-125b-5p mimics. They turned into large, cytoplasm-rich macrophages, indicating a more active status, whereas the control group showed more spindle-shaped cells (Figure 5B). Cell-size quantification also showed a significantly increased cell area for miR-125b-5p-transfected macrophages (Figure 5C). Thus, overexpression of miR-125b-5p reflected the phenotypic switch of M1 macrophages induced by melanoma-derived exosomes.

To investigate a more functional parameter we overexpressed the miR-125b-5p mimic or control mimic in M1 polarized macrophages and analyzed apoptosis and cell viability in vitro. Flow cytometry showed a less spontaneous apoptosis of macrophages when overexpressing miR-125b-5p (Figure 5D). Consistently cell count analysis showed a significantly higher number of macrophages when transfected with miR-125b-5p (Figure 5E). Using viability staining assay, we recorded more viable macrophages when transfected with the miR-125b-5p mimic compared to control mimics after 96 h indirect co-culture with melanoma cells (MV3) (Figure 5F). From these results, we conclude that miR-125b-5p promotes the survival of macrophages.

### 2.6. miR-125b-5p Targets Lysosomal Acid Lipase A (LIPA)

miRNAs regulate the function and phenotype of cells by post-transcriptional regulation. In order to find potential miR-125b-5p target mRNAs, we performed computational target prediction (TargetScan and miRTarBase) and compared it to the expression profile of exosome-treated M1 macrophages. Here we found 19 putative miR-125b-5p targets downregulated in M1 macrophages treated with melanoma-derived exosomes or co-cultured with melanoma cells (Figure 6A). The most abundant and one of the strongest repressed transcripts in exosome-treated M1 macrophages was the lysosomal acid lipase A (LIPA) (Figure 6B). M1 macrophages treated with exosomes showed a decreased expression of genes involved in lipid catabolic processes, including LIPA (Figure 6C). Because others have already described that LIPA deficiency induces a proinflammatory phenotype in macrophages and hepatocytes [27,28,29,30], we analyzed if miR-125b-5p represses LIPA. The overexpression of miR-125b-5p mimics in M1 polarized macrophages strongly reduces LIPA protein compared to M1 macrophages transfected with control mimics (Figure 6D). As miR-125b-5p has been demonstrated to bind directly to the LIPA 3’UTR [31], our data indicates that melanoma exosome-delivered miR-125b-5p directly targets LIPA in TAMs, which in turn induces a proinflammatory and tumor-permissive phenotype.

## 3. Discussion

In this study, we report that exosomes derived from melanoma cells show a specific miRNA profile and induce a tumor-promoting phenotype in macrophages. Especially miR-125b-5p contributes to tumor-permissive properties and the prolonged survival of macrophages. Its effect is at least partially mediated by its targeting of LIPA.

We demonstrated that THP-1-derived macrophages take up exosomes secreted by melanoma cells, thereby inducing a phenotype with features of both M1 and M2 macrophages. Transcriptomic pathway analysis showed an enrichment of the M1 marker genes associated with inflammation, angiogenesis, and chemotaxis, as well as IL-10 and IL-4/IL-13 signaling, which are M2-associated pathways. Similar results have been observed with exosomes from pancreatic cancer cells and melanoma [6,32]. However, these studies did not investigate the exosome cargo and molecular mechanisms for the phenotypical switch of macrophages. Nevertheless, these results comply with the observed plasticity of TAMs in vivo, which does not reflect a rigid, robust distinction into classical M1 and alternative M2 polarized macrophage phenotypes in vitro [17]. Rather, TAMs often express high amounts of proinflammatory, proangiogenic, and chemotaxis-inducing cytokines and chemokines and may, in this way, facilitate tumor growth, progression, and metastasis [13,14].

miRNAs are the most potent candidates to mediate the effects of exosomes, because a single miRNA can regulate the expression of several proteins. Therefore, we analyzed the miRNA load of melanoma-derived exosomes to investigate the molecular mechanisms behind the observed macrophage reprogramming. We identified miR-125b-5p as one of the most abundant and significant enriched miRNAs in exosomes secreted by melanoma cells compared with NHEM-derived exosomes. Correspondingly, we demonstrated an enrichment of the mature miR-125b-5p in M1 macrophages previously co-cultured with melanoma cells or treated with melanoma-derived exosomes. In contrast, the pri-miR-125b-1 or pri-miR-125b-2 expression was not changed, which indicates that the enrichment was due to exosome delivery rather than caused by increased endogenous or exosome-triggered expression.

In a study investigating the influence of tumor-derived microRNAs on the induction of MDSCs and their relevance in the prediction of the resistance of immunotherapy, miR-125b-5p was detected in melanoma cell-derived microvesicles, without addressing its target or its molecular function [33].

With regard to its possible function, the expression of miR-125b-5p was shown to be increased in primary melanoma in comparison to metastasis [34,35,36]. Apart from its expression in melanoma cells, miR-125b-5p was reported to induce the adaption to inflammation, recruitment, and activation of myeloid cells [37,38,39,40,41], while it represses the cytotoxicity of lymphoid [42] immune cells. We showed that treatment with melanoma-derived exosomes or miR-125b-5p overexpression induced a morphological switch of M1 macrophages together with an enforced expression of IL-1β, CCL-1, CCL2, and CD80 (B7-1). Our observations are supported by previous reports, showing that overexpression of miR-125b-5p potentiates macrophage activation by the induction of CD80 and TNF expression, as well as changes in morphology [37,38].

These results lead to the hypothesis that miR-125b-5p is a central regulator in early melanoma establishment. As we showed, melanoma exosome-delivered miR-125b-5p reinforces the activation of M1 macrophages through the induction of CCL1, CCL2, and IL-1β expression, thus contributing to myeloid cell recruitment and cancer-associated inflammation. In addition, the CD80 expression induced by miR-125b bore the potential to influence the T cell-mediated specific immune response. The co-stimulatory factor CD80 was shown to have an increased binding affinity to the immune checkpoint molecule CTLA4 [43], which is expressed on activated T cells. Increased CD80 expression on melanoma-exosome-primed macrophages may lead to an inhibition of activated T-cell proliferation and function.

miR-125b-5p controls monocytes’ adaption to inflammation [37] and promotes the survival of myeloid, cancer, and endothelial cells [39,44,45]. We found that miR-125b-5p overexpression represses spontaneous apoptosis and promotes the survival of M1 polarized macrophages with and without melanoma cell co-culture.

From all the putative targets of miR-125b-5p, we identified LIPA as the most abundant and strongly decreased transcript following treatment of M1 macrophages with melanoma-derived exosomes. LIPA is a crucial regulator of lysosomal lipolysis and essential for cell fatty acid metabolism. LIPA hydrolyzes cholesteryl esters and triglycerides to generate free fatty acids as fuel for fatty-acid oxidation [46]. While inflammatory M1 macrophages rely on glycolysis, fatty-acid oxidation is significant for the polarization of anti-inflammatory M2 macrophages [47]. Depletion of LIPA or inhibition of fatty-acid oxidation disrupts M2 activation pathways [47]. We showed that the expression of lipid metabolism-associated genes, including LIPA, were strongly decreased in M1 macrophages previously treated with melanoma-derived exosomes. Additionally, we showed that the overexpression of miR-125b-5p in M1 polarized macrophages abolished LIPA protein. As LIPA was shown to be a direct target of miR-125b-5p [48], we postulate that exosome-delivered miR-125b-5p represses LIPA in M1 macrophages. Consistently with our findings, LIPA deficiency and miR-125 overexpression in macrophages both induce identical pathways regulating proinflammatory factors and myeloid cell recruitment [29,30,49,50]. Correspondingly, LIPA knockout in melanoma-associated myeloid cells promotes cancer cell proliferation and metastasis [51].

## 4. Materials and Methods

### 4.1. Cell Cultures

Melanoma cell lines (MV3, WM9, WM902B, and WM35) were cultured in DMEM, and THP-1 cells were cultured in RPMI-1640, all supplemented with 10% fetal calve serum (FCS) and 1% penicillin-streptomycin. Primary human melanocytes were cultured in medium 254 (Cascade Biologics^®^) including human melanocyte growth supplement (HMGS) and 1% penicillin-streptomycin. All cells were incubated at 37 °C and 5% CO_2_.

### 4.2. Exosome Isolation, Analysis, and Staining

Cells were cultured for 48 h or 72 h in DMEM supplemented with 10% exosome-depleted FCS (Thermo Fisher Scientific, Waltham, MA, USA) and 1% penicillin-streptomycin. Supernatant was collected and centrifuged for 5 min at 1000 rpm to remove cells and cell debris, followed by 30 min at 6000 rpm to remove larger vesicles. Afterwards, the supernatant was filtered through a 0.2 µm filter and centrifuged at 100,000× *g* for 2 h. Pelleted exosomes were washed with PBS and centrifuged for another 2 h at 100,000× *g*. Exosomes were resuspended in PBS. Alternatively, exosomes were isolated using an exosome isolation reagent for cell culture (Thermo Fisher Scientific), following the manufacturer’s instructions. Exosome analysis was performed by nanoparticle tracking analysis (NTA) using a NanoSight NS300 (Malvern Panalytical, Kassel, Germany). Isolated exosomes were stained with SYTO^®^ RNASelect™ (Thermo Fisher Scientific) regarding the manufacturer’s instructions and analyzed by fluorescence microscopy and flow cytometry.

### 4.3. THP-1 Macrophage Polarization

For macrophage polarization, THP-1 cells were stimulated with 150 nM phorbol 12-myristate 13-acetate (PMA) for 24 h to get Mφ macrophages. Afterwards, Mφ rested for 24 h in RPMI-1640 containing 10% FCS and 1% penicillin-streptomycin. For the stimulation of the M1 macrophages, Mφ were cultured in RPMI-1640 containing 10% FCS, 1% penicillin-streptomycin, LPS (10 pg/mL), and INFγ (20 ng/mL) for 24 h. For the polarization of the M2 macrophages, Mφ were cultured in RPMI-1640 containing 10% FCS, 1% penicillin-streptomycin IL-4 (20 ng/mL), and IL13 (20 ng/mL) for 24 h.

### 4.4. Transcriptome Analysis

50–100 ng of the total RNA were fragmented by the addition of a 5x fragmentation buffer (200 mM Tris acetate, pH 8.2, 500 mM potassium acetate, and 150 mM magnesium acetate) and heating at 94 °C for 2 min in a thermocycler followed by ethanol precipitation with ammonium acetate and GlycoBlue (Life Technologies, Carlsbad, CA, USA) as the carrier. The fragmented RNA was further processed using the Ovation Human FFPE RNA-Seq Library Systems (Nugen, Redwood City, CA, USA) according to the instructions of the manufacturer. This library preparation included reverse transcription using random priming, second strand synthesis, blunt-end repair, adapter ligation with nucleotide analog-marked adaptors, strand selection, and insert dependent adapter cleavage (inDA-C) to remove rRNA, globin, and other house-keeping transcripts. Target sequences for inDA-C were derived from the human genome. The barcoded libraries were purified and quantified using the Library Quantification Kit—Illumina/Universal (KAPA Biosystems). A pool of libraries was used for sequencing at a concentration of 10 nM. Sequencing of 2 × 150 bp was performed with an Illumina NextSeq 550 sequencer at the sequencing core facility of the IZKF Leipzig (Faculty of Medicine, University Leipzig, Leipzig, Germany) according to the instructions of the manufacturer. Demultiplexing of raw reads, adapter trimming, and quality filtering was done according to Stokowy and colleagues (27). Mapping against the human reference genome (hg38) of polished reads was done using HISAT2, Stringtie, and the R package Ballgown (28).

### 4.5. Small RNA-Seq Protocol

10–50 ng of the total RNA was used in the small RNA protocol with the NEXTflex Small RNA-seq Kit v3 (Bioo Scientific, Austin, TX, USA) according to the instructions of the manufacturer. A pool of libraries was used for sequencing at a concentration of 10 nM. Sequencing of 1 × 75 bp was performed with an Illumina NextSeq 550 sequencer at the sequencing core facility of the IZKF Leipzig (Faculty of Medicine, University Leipzig) according to the instructions of the manufacturer. Demultiplexing of raw reads, adapter trimming, and quality filtering was done according to Stokowy et al. (27), using the adapter sequences of the NEXTflex kit containing random bases next to the library insert. Mapping against the human reference genome (hg38) and miRbase reference sequences (v22) was done using Bowtie2 (29). Read counts were calculated with the R Bioconductor package Rsamtools (http://bioconductor.org/packages/release/bioc/html/Rsamtools.html) and normalized using the DESeq2 (30) and EdgeR (31) R Bioconductor packages.

### 4.6. miRNA Detection by Quantitative Real-Time PCR

The total RNA was extracted from cells or exosomes using the TriFast™ reagent (Peqlab). The miRNA quantification was performed by qRT-PCR using TaqMan^®^ MicroRNA Reverse Transcription Kit and TaqMan^®^ Universal Master Mix II following the manufacturer’s instructions. For normalization, RNUB6 or cel-miR-39 expression was used. TaqMan^®^ reverse transcription (RT) and PCR primers for RNUB6 and hsa-miR-125b-5p were obtained from Thermo Fisher Scientific. For the analysis of pri-miR-125b-1 and pri-miR-125b-2 we used pri-miR assays from Thermo Fisher Scientific, and the samples were normalized by GAPDH.

### 4.7. Transcriptional Analysis by qRT-PCR

The total RNA was isolated using TriFast™ reagent (Peqlab). For quantitative RT-PCR we used PowerUp™ SYBR™Green master mix (Thermo Fisher Scientific) following the manufacturer’s instructions. qPCRs were performed on QuantStudio™ 5 Real-Time PCR Systems. For normalization we used GAPDH. The following primers were used:CCL1 for   GCTTCACCAGGCTCATCAAACCL1 rev   TCAGGGGAATCTCTTGCTCCCCL2 for   CTCTCGCCTCCAGCATGAAACCL2 rev   CTTGAAGATCACAGCTTCTTTGGCD80 for   GGGAAATGTCGCCTCTCTGACD80 rev   TGCTCACGTAGAAGACCCTCGAPDH for ACCACAGTCCATGCCATCACGAPDH rev TCCACCACCCTGTTGCTGTAIL1b for TGATGGCTTATTACAGTGGCAATGIL1b rev GTAGTGGTGGTGGGAGATTCGIL6 for GACAGCCACTCACCTCTTCAGAIL6 rev GTGCCTCTTTGCTGCTTTCACIL8 for GCTAAAGAACTTCGATGTCAGTGCIL8 rev CTCAGCCCTCTTCAAAAACTTCTCTNFA for  CTTCTGCCTGCTGCACTTTGTNFA rev  GGCCAGAGGGCTGATTAGAGAVEGFA for CACCATGCAGATTATGCGGAVEGFA rev GAGGCTCCAGGGCATTAGAC

### 4.8. Transfections

miR-125b-5p mimics and the negative control were purchased by Qiagen (Hilden, Germany). Cells were transfected with 100 nM mimics using Lipofectamine 3000 reagent (Invitrogen) following the manufacturer’s instructions.

### 4.9. Immunoblot Analyses

Cells and exosomes were lysed by a RIPA buffer for 30 min at 4°C. Protein extracts were resolved by SDS–PAGE and blotted to nitrocellulose membranes and probed with the following antibodies: anti-GAPDH, anti-CD80, anti-CD63, and anti-calnexin (Santa Cruz, Dallas, TX, USA), as well as anti-lysosomal acid lipase/LAL (Abcam, Cambridge, UK). For antibody detection we used anti-rabbit IgG-HRP (Cell Signaling Technology, Danvers, MA, USA) or m-IgGκ BP-HRP (Santa Cruz, Dallas, TX, USA).

### 4.10. Apoptosis Assay

For apoptosis analysis we used the Annexin V Staining Kit (PE; 7AAD) (BD Biosciences, Franklin Lakes, NJ, USA) following the manufacturer’s instructions. Subsequently, the cells were washed in phosphate-buffered saline and analyzed with a BD FACS Scan cytometer using CellQuest software (BD Biosciences, Franklin Lakes, NJ, USA).

### 4.11. Viability Staining Assay

Cells were washed with PBS and fixed for 20 min with methanol. Then cells were stained with a gentian-violet stain for 15 min and washed twice with water. Afterwards, cells were documented by microscopy. Stained cells were incubated with 0.1 M Na citrate for 20 min to resolve the staining. For quantification, supernatant absorption was measured at 620 nm with the Tecan multi plate reader infinite m200 pro.

### 4.12. miRNA Pathway Analysis

For computational exosome miRNA pathway analyses we used miRNet [52,53]. Pathway analyses were performed for the top five abundant and significantly enriched miRNAs in melanoma exosomes (hsa-miR-100-5p, hsa-miR-99b-5p, hsa-miR-221-3p, hsa-miR-24-3p, and hsa-miR-125b-5p) by the default settings, using Kyoto Encyclopedia of Genes and Genomes (KEGG) pathway analysis.

### 4.13. Statistical Analyses

We used the Student’s t-test to determine the statistical significance of experimental results for experiments with two groups. For experiments with multiple groups statistical analysis was performed using one-way ANOVA, followed by Dunnett’s comparisons test against the control group. A p-value of 0.05 or less was considered significant. The results were represented as the average ± standard deviation from at least three independent experiments.

## 5. Conclusions

In summary, we conclude that exosomes derived from melanoma induce a tumor-promoting phenotype in macrophages. Exosome-delivered miR-125b-5p contributes to this macrophage phenotype switch by targeting LIPA 3’UTR.

According to its gene expression profile the macrophage phenotype generated by melanoma-derived exosomes or miR-125b-5p has the potential to induce tumor-associated inflammation and angiogenesis as well as myeloid cell recruitment and survival.

## Figures and Tables

**Figure 1 cancers-12-00464-f001:**
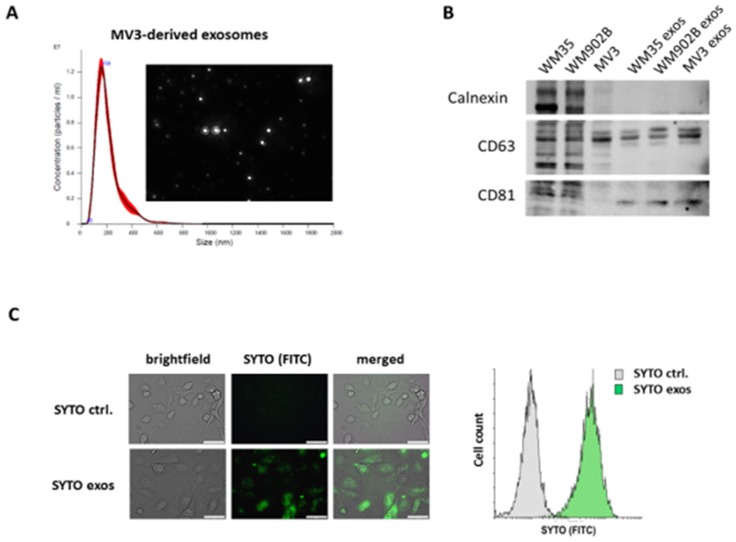
Isolation and characterization of exosomes. Exosomes were isolated by ultracentrifugation. (**A**) Exemplary nanoparticle tracking analysis (NTA) of isolated exosomes showed a size distribution of approximately 160 nm. (**B**) Western blot analysis of whole cell lysates and lysates of isolated exosomes for CD81, CD63, and calnexin. (**C**) Representative microscopy of macrophages incubated for 24h with SYTO-stained exosomes or PBS as control (scale bar: 50 µm). Flow cytometric analyses demonstrate the exosome uptake by THP-1-derived exosomes.

**Figure 2 cancers-12-00464-f002:**
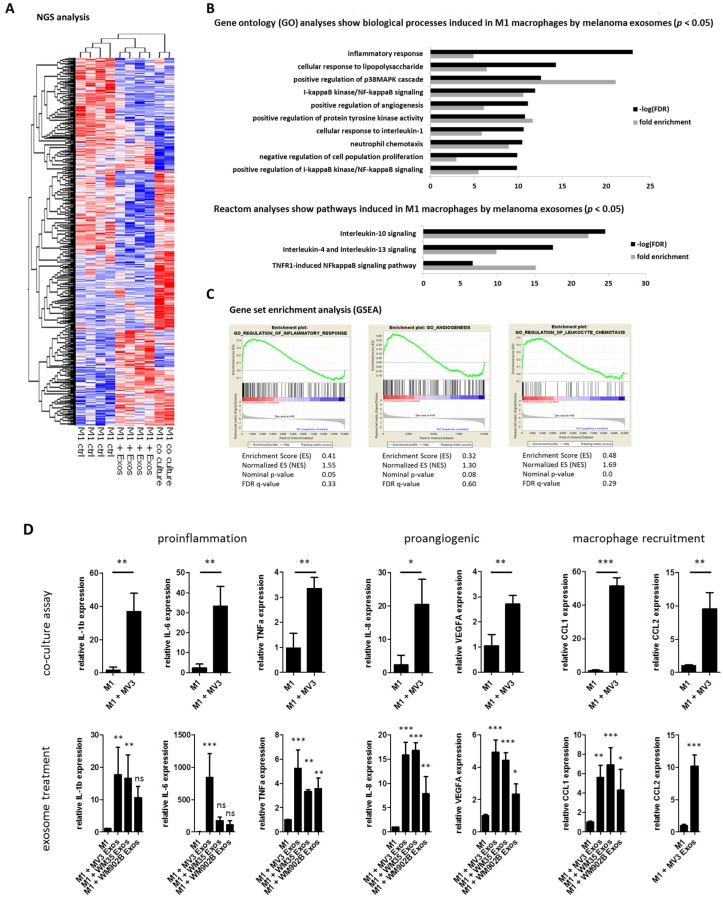
Melanoma exosomes induce a proinflammatory and proangiogenic phenotype macrophages. (**A**) Hierarchical clustering of differential transcripts of THP-1-derived M1 macrophages 48 h co-cultured with MV3 melanoma cells, treated with MV3 released exosomes or M1 macrophages alone. (**B**) Gene ontology and Reactome pathway analysis of genes significantly increased in M1 macrophages after treatment with melanoma-derived exosomes. (**C**) Gene set enrichment analysis (M1 + exosomes versus M1 control) of next-generation sequencing results shows induction of inflammation, angiogenesis, and leukocyte chemotaxis. (**D**) Quantitative RT-PCR validated induction of proinflammatory, proangiogenic, and chemotaxis-related gene expression after 48 h co-culture of M1 with MV3 cell line or treatment with melanoma cell-derived exosomes. Bars represent the average ± standard deviation of at least three independent experiments (* *p* ≤ 0.05; ** *p* ≤ 0.01; *** *p* ≤ 0.001; ns: not significant). (**E**) Bright-field microscopy shows representative morphological changes (red arrows) of THP-1-derived M1 macrophages with and without MV3- derived exosomes treatment (48 h). White scale bar: 200 µm; black scale bar: 50 µm.

**Figure 3 cancers-12-00464-f003:**
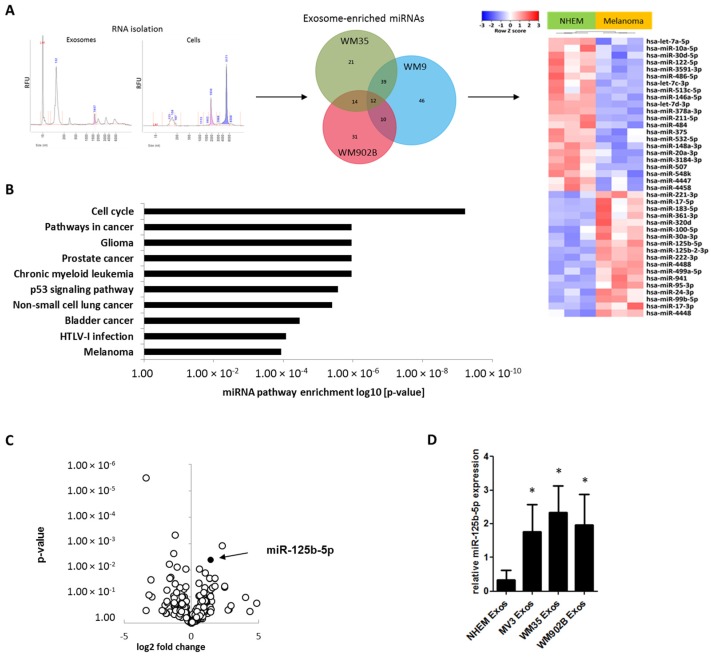
miR-125b-5p is enriched in melanoma exosomes. Next-generation sequencing (NGS) analysis of exosome small RNA load. (**A**) Quality control of isolated RNA from exosomes and whole cells. Heatmap shows the differential miRNA expression of normal human epidermal melanocytes (NHEMs) and melanoma cell lines (WM9, WM35, and WM902B). Venn diagram shows differential enrichment of miRNAs in exosomes of different cell lines compared to NHEMs. (**B**) miRNet miRNA pathway enrichment of the top five abundant and most significantly enriched miRNAs in the melanoma-derived exosomes (hsa-miR-100-5p, hsa-miR-99b-5p, hsa-miR-221-3p, hsa-miR-24-3p, and hsa-miR-125b-5p). (**C**) Volcano plot shows miRNA enrichment of miR-125b-5p in exosomes derived from the melanoma cell lines compared (WM9, WM35, and WM902B) to NHEM-derived exosomes. (**D**) Validation of miR-125b-5p enrichment in melanoma cell line-derived exosomes by qRT-PCR. Bars represent the average ± standard deviation of at least three independent experiments (* *p* ≤ 0.05).

**Figure 4 cancers-12-00464-f004:**
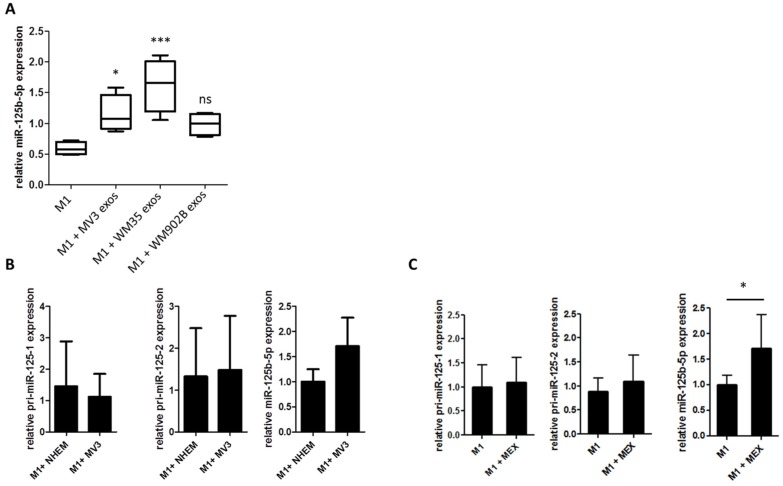
miR-125b-5p is delivered into M1 macrophages by melanoma-derived exosomes. (**A**) qRT-PCR for miR-125b-5p in THP-1-derived M1 macrophages with and without 48 h treatment with exosomes derived from different melanoma cell lines (* *p* ≤ 0.05; *** *p* ≤ 0.001; ns: not significant). qRT-PCR analysis of primary and mature transcripts of miR-125b after (**B**) co-culture (48 h) of THP-1- derived M1 macrophages with NHEM or MV3 melanoma cell line or (**C**) treated (48 h) with and without MV3-derived exosomes. Bars represent the average ± standard deviation of at least three independent experiments (* *p* ≤ 0.05).

**Figure 5 cancers-12-00464-f005:**
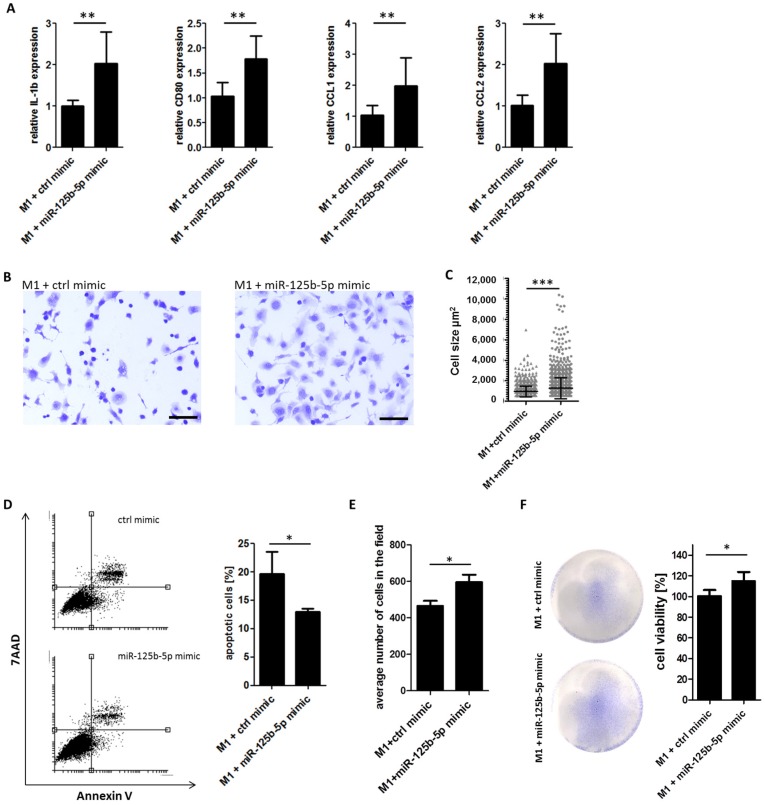
Overexpression of miR-125b-5p partially mimics a exosome-induced TAM phenotype switch and promotes the survival of macrophages. THP-1-derived M1 macrophages were transfected for 48 h with miR-125b-5p mimics (100 nM) or a control mimic (100 nM), respectively. (**A**) qRT-PCR analyses show an induction of IL-1β, CD80, CCL1, and CCL2 after miR-125b-5p overexpression. Bars represent the average ± standard deviation of at least three independent experiments (** *p* ≤ 0.01). (**B**) Representative pictures of fixed and crystal violet-stained M1 macrophages show a morphological switch by miR-125b-5p overexpression. Scale bar: 100µm. (**C**) Quantification of the cell size of M1 macrophages 48 h after transfection with the control mimic or miR-125b-5p mimic (*** *p* ≤ 0.001). (**D**) Flow cytometry analysis of apoptosis 48 h after transfection. Bars represent the average ± standard deviation of at least three independent experiments (* *p* ≤ 0.05). (**E**) Cell count analysis of M1 polarized macrophages 48 h after transfection with the control mimic or miR-125b-5p mimic. Bars represent the average ± standard deviation of at least three independent experiments (* *p* ≤ 0.05). (**F**) Transfected macrophages were 72 h co-cultured indirectly with the MV3 melanoma cell line. Macrophages were fixed and stained with gentian violet viability stain. Representative pictures show stained M1 macrophages in a culture dish. Viability of macrophages was quantified measuring absorption at 620 nm by a multi plate reader. Bars show the average ± standard deviation of at least three independent experiments (* *p* ≤ 0.05).

**Figure 6 cancers-12-00464-f006:**
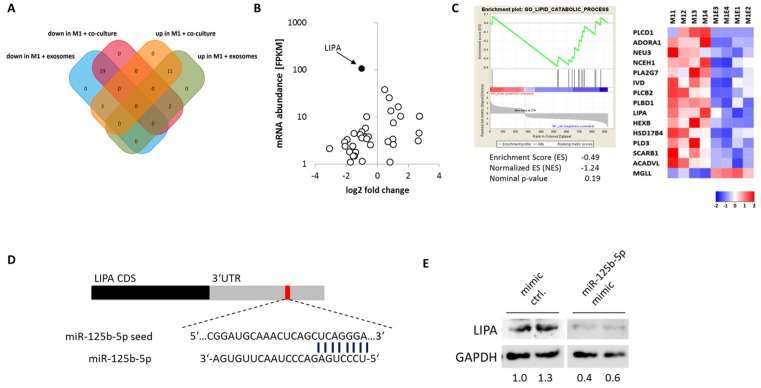
miR-125b-5p targets LIPA. (**A**) Venn diagram shows expression (NGS) direction overlap of potential miR-125b-5p target mRNAs in M1 macrophages after 48 h co-culture with MV3 cells or MV3-derived exosome treatment. (**B**) Volcano plot examines the mRNA abundance [FPKM] and log2 fold change (M1 + Exos versus M1; *p* ≤ 0.05) of predicted miR-125b-5p targets. (**C**) Gene set enrichment analysis shows a decreased expression of genes associated with lipid metabolism after M1 macrophages were treated (48 h) with melanoma exosomes. Heatmap shows differential expression of gene set: GO lipid catabolic process. (**D**) Scheme of predicted miR-125b-5p binding site in lysosomal acid lipase A (LIPA) 3’UTR. (**E**) Western blot for LIPA protein expression in M1 macrophages 48 h after transfection with miR-125b-5p mimic or control mimic, respectively.

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
