# Peer review of "Melanoma-Derived Exosomal miR-125b-5p Educates Tumor Associated Macrophages (TAMs) by Targeting Lysosomal Acid Lipase A (LIPA)"

_cancers, 2020, doi:10.3390/cancers12020464_

Round 1
Reviewer 1 Report
This study evaluates the effect of exosomes derived from well-established cutaneous melanoma cell lines on phenotypic switch of tumor associated macrophages. The authors first demonstrate that melanoma derived exosomes trigger morphological changes and upregulation of proinflammatory as well as proangiogenic genes in macrophages. They further evaluated the miR content of melanoma derived exosomes and identified an enrichment of miR-125b-5p. Upregulation of miR-125b-5p in macrophages results in upregulation of Il-1β, CCL1, CCL2 and CD80 but not IL-6, TNFα, VEGFA and IL8. miR-125b-5p upregulation also increased cellular size and decreased the number of apoptotic cells. The authors further identified Lysosomal acid lipase A as a target of miR-125b-5p.
This is an interesting and well-conducted study assessing the effects of melanoma derived exosomes on phenotypic switch of TAM.
A few comments:
TAM can also produce exosomes that can affect the local microenvironment. Although the authors observed different effects in macrophages with or without melanoma derived exosomes, have they tried to assess the presence of exosomes in the supernatant of macrophages alone?
It is a detail, but it would be useful for the reader to know directly in the abstract that exosomes were derived from cutaneous melanoma and not from melanoma from other origin.
In my opinion, the first paragraph of the introduction is not necessary.
Author Response
TAM can also produce exosomes that can affect the local microenvironment. Although the authors observed different effects in macrophages with or without melanoma derived exosomes, have they tried to assess the presence of exosomes in the supernatant of macrophages alone?
Answer: Yes, we would also expect that TAMs influence the tumor microenvironment by exosomes. However, we did not yet analyze exosomes derived by macrophages or by melanoma exosome treated macrophages, but it could be very interessting too.
It is a detail, but it would be useful for the reader to know directly in the abstract that exosomes were derived from cutaneous melanoma and not from melanoma from other origin.
Answer: We specified cutaneous melanoma in the abstract.
In my opinion, the first paragraph of the introduction is not necessary.
Answer: We excluded the first paragraph of the introduction.
Reviewer 2 Report
This paper was a pleasure to read. The authors showed that miRNA containing exosomes from melanoma cells were able to change macrophage phenotype. They identified miRNAs in the exosomes and showed that the most abundant miRNA was capable of eliciting at least some of the phenotypic changes observed in the macrophages. The data was clear and well presented.
1) Some of the experiments had more than two groups so an ANOVA followed by post-hoc analysis is more appropriate for determining statistical significance than a Students T-test.
Author Response
Some of the experiments had more than two groups so an ANOVA followed by post-hoc analysis is more appropriate for determining statistical significance than a Students T-test.
Answer: For the experiments with multiple groups we skipped the t-test and performed ANOVA followed by Dunnett’s post-hoc analysis to compare our treated groups to the control group.